# UNIFIED VISION-LANGUAGE-ACTION MODEL

**Yuqi Wang**[1][*]   **Xinghang Li**[2]   **Wenxuan Wang**[1,2]   **Junbo Zhang**[3]   **Yingyan Li**[1]
**Yuntao Chen**[4]   **Xinlong Wang**[2][✉]   **Zhaoxiang Zhang**[1][✉]

[1] Institute of Automation, Chinese Academy of Sciences (CASIA)

[2] Beijing Academy of Artificial Intelligence (BAAI)   [3] Tsinghua University

[4] Centre for Artificial Intelligence and Robotics, Hong Kong Institute of Science & Innovation

Project page: https://robertwyq.github.io/univla.github.io

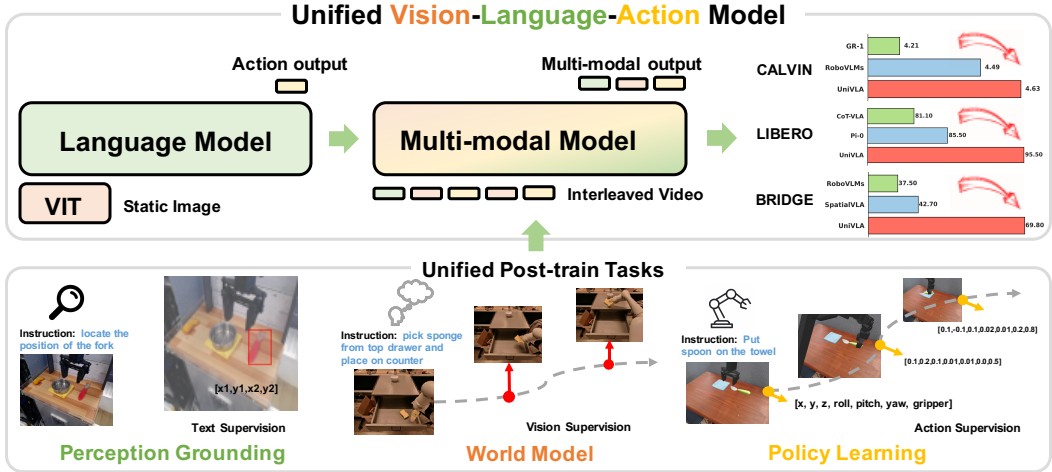

Figure 1: **We present UniVLA, a unified vision-language-action model.** Unlike prior VLA approaches that typically rely on an extra vision encoder to extract image features and generate only action outputs, UniVLA represents vision, language, and action as discrete tokens within a unified autoregressive framework. This unified modeling paradigm enables multi-modal outputs and supports a wide range of tasks—such as text-supervised perception grounding, vision-supervised world modeling, and action-supervised policy learning—within a single architecture. The unified token-based design further allows UniVLA to effectively leverage large-scale multimodal data, particularly video, for scalable and generalizable learning. UniVLA achieves new state-of-the-art results on CALVIN, LIBERO, and SimplerEnv-Bridge, significantly surpassing existing methods.

## ABSTRACT

Vision-language-action models (VLAs) have garnered significant attention for their potential in advancing robotic manipulation. However, previous approaches predominantly rely on the general comprehension capabilities of vision-language models (VLMs) to generate action signals, often overlooking the rich temporal and causal structure embedded in visual observations. In this paper, we present UniVLA, a unified and native multimodal VLA model that autoregressively models vision, language, and action signals as discrete token sequences. This tokenized formulation naturally supports flexible multimodal task learning, particularly from large-scale video data, and further demonstrates that generative vision supervision can significantly enhance visual understanding. By incorporating world modeling during post-training, UniVLA captures causal dynamics from videos, facilitating effective transfer to downstream policy learning—especially for long-horizon tasks. Our approach sets new state-of-the-art results across several widely used simulation benchmarks, including CALVIN, LIBERO, and Simplenv-Bridge, substantially outperforming prior methods. For example, UniVLA achieves 95.5% average success rate on LIBERO benchmark, surpassing $\pi_0$-FAST's 85.5%. We further

[*]Work done during an internship at BAAI. [✉] Corresponding author

demonstrate its broad applicability through experiments on real-world ALOHA manipulation tasks and autonomous driving scenarios.

# 1 INTRODUCTION

Developing agents capable of perceiving, reasoning, and acting in the physical world has long been a central objective of artificial intelligence. Recent advances in vision-language-action (VLA) models Brohan et al. (2023); Octo Model Team et al. (2024); Kim et al. (2024); Black et al. (2024), grounded in the powerful generalization capabilities of vision-language models (VLMs) Peng et al. (2023); Jaech et al. (2024); Beyer et al. (2024); Wang et al. (2024a); Guo et al. (2025), have demonstrated impressive performance across a wide range of robotic manipulation tasks, and are increasingly being adapted to generalist humanoid robots Bjorck et al. (2025); Ding et al. (2025) that demand broader embodied intelligence. However, most existing VLA approaches Kim et al. (2024); Black et al. (2024) follow a language-centric paradigm: visual observations are first projected into a semantic space, and action policies are subsequently derived based on these representations. This late-fusion strategy, while beneficial for semantic understanding and generalization, limits the formation of deeply coupled cross-modal representations and impedes the learning of temporal and causal dependencies across the perception-action loop. This raises a central question: *Can vision, language, and action be jointly modeled within a unified representation space to facilitate tighter cross-modal integration and more effective policy learning?*

While appealing in theory, unified modeling presents two key challenges. First, vision, language, and action are inherently heterogeneous modalities: vision comprises high-dimensional, continuous spatial signals; language conveys abstract, discrete semantics; and actions involve temporally ordered sequences with causal dependencies. Second, the perception-to-action pipeline is inherently dynamic and causal, yet existing VLA models Brohan et al. (2023); Kim et al. (2024); Black et al. (2024) often adopt static, language-centric paradigms that merely learn the mapping from static image to action. These models fail to capture the dynamic nature of real-world interactions, thereby limiting their ability to leverage the rich temporal information from videos for training.

To address the above challenges, we introduce UniVLA, a novel framework for unified vision–language–action learning. As illustrated in Figure 1, we propose a unified framework that supports both *multimodal* and *multi-task* learning. At the modality level, vision, language, and action signals are all transformed into discrete tokens and modeled using a shared vocabulary. This unified token representation allows for joint learning across modalities, fostering deeper cross-modal understanding and integration. Building upon the unified framework, we adopt an autoregressive, Markov chain-based sequence modeling approach, where observations and actions are interleaved. This structure naturally incorporates causal dependencies, enabling the model to reason over temporal dynamics rather than treating perception and action as isolated tasks. By integrating the world model paradigm during training, we leverage large-scale robotics videos for self-supervised learning, allowing the model to capture environment dynamics in a temporally consistent and causally grounded manner. Remarkably, we find that post-training with world models significantly enhances policy learning, particularly for long-horizon and out-of-distribution tasks.

Experiments across multiple simulation benchmarks, including CALVIN Mees et al. (2022b), LIBERO Liu et al. (2023), and SimplerEnv Li et al. (2024d), demonstrating clear performance improvements over existing methods. Our model incorporates world model learning during post-training, enabling it to effectively capture visual dynamics from large-scale videos. This strategy significantly enhances both data efficiency and training efficiency in downstream policy learning, and allows for rapid adaptation to novel robotic tasks. Beyond policy learning, we demonstrate the model's multimodal output capabilities, including spatial reasoning and visual prediction, highlighting its versatility. Furthermore, we extend our approach to autonomous driving scenarios for broader applicability. These results underscore the potential of our unified VLA model as an alternative and promising direction for generalist embodied intelligence.

Our contributions are summarized as follows:

- We propose UniVLA, the first unified vision–language–action (VLA) model that encodes vision, language, and action as discrete tokens within a shared vocabulary, jointly modeling them through autoregressive sequence learning. This approach offers a novel architecture

alternative to the existing VLA paradigm, facilitating more integrated cross-modal modeling and enabling large-scale video-based training.

- Our unified sequence modeling framework supports a broad range of multimodal tasks. Through extensive experiments with various post-training strategies, we show that world models can effectively capture temporal dynamics from video data, substantially boosting performance and improving both data and training efficiency in downstream policy learning—spanning simulation benchmarks, real-world robotic platforms, and even driving domains, where world model post-training consistently benefits policy learning.

- Our model achieves state-of-the-art performance across multiple simulated benchmarks (CALVIN, LIBERO, and SimplerEnv-Bridge) and introduces an open-source VLA framework enabling large-scale video training. We further demonstrate strong multimodal capabilities, including spatial reasoning and video prediction, and validate effective transfer to real-world ALOHA platforms and autonomous driving scenarios, highlighting its promise for generalist embodied intelligence.

## 2    RELATED WORKS

### 2.1    VISION-LANGUAGE-ACTION MODELS

Recent vision-language-action (VLA) models have demonstrated strong task performance across diverse robots and tasks Brohan et al. (2023); Vuong et al. (2023); Driess et al. (2023); Kim et al. (2024); Zhen et al. (2024); Cheang et al. (2024); Black et al. (2024); Zheng et al. (2024); Liu et al. (2025); Kim et al. (2025); Intelligence et al. (2025). These models leverage pre-trained vision-language models (VLMs) to enhance understanding and generalization, further fine-tuned on large-scale robotic datasets for low-level control. Currently, VLA models can be categorized into two paradigms based on their output space: *pure action prediction* and *visual-guided action prediction*.

**Pure action prediction.** Recent efforts have extended vision-language models (VLMs) to incorporate action modalities, enabling direct action prediction from visual and language inputs. A prominent example is RT-2Brohan et al. (2023), which learns from both internet-scale and robotic data to generate discrete actions autoregressively, showcasing strong generalization and semantic grounding. Building upon this, RT-HBelkhale et al. (2024) introduces hierarchical actions to facilitate data sharing across tasks. OpenVLAKim et al. (2024) scales this paradigm with a 7B-parameter open-source model trained on 970k real-world demonstrations spanning diverse datasets. To enhance spatial reasoning, SpatialVLAQu et al. (2025) integrates spatial representations into the action modeling process. Beyond architecture scaling, new action modeling techniques have also emerged. $\pi_0$ Black et al. (2024) incorporates flow matching to improve action learning efficiency, while FAST Pertsch et al. (2025) introduces a unified frequency-domain formulation for discretizing actions.

**Visual-guided action prediction.** These studies leverage the power of visual pretraining, typically based on a policy-as-video formulation, by predicting future visual signals and subsequently decoding them into actions. SuSIE Black et al. (2023) predicts key future frames and derives actions through inverse dynamics. UniPi Du et al. (2023) generates videos from text instructions, extracting actions from the frames. GR series Wu et al. (2024); Cheang et al. (2024); Li et al. (2025a) leverages video pretraining for general policy learning. PAD Guo et al. (2024) uses diffusion models to simultaneously learn future images and actions. LAPA Ye et al. (2025) proposes to learn latent actions between images with VQ-VAE from action-free internet-scale videos. Track2Act Bharadhwaj et al. (2024) extracts point tracks from diverse web videos to guide the learning of interaction plans.

Both approaches have complementary strengths and limitations. The first, focused on action prediction, integrates naturally with Vision-Language Models (VLMs) but lacks spatial reasoning and visual forecasting. The second, incorporating visual generation, requires separating generative and action prediction models, limiting unified multimodal learning. Our work bridges these paradigms by combining video-centric pretraining with VLM capabilities, resulting in a native multimodal model with strong future potential.

Regarding comparisons with current VLA architectures such as $\pi_{0.5}$ Intelligence et al. (2025), our framework prioritizes video-centric modeling during the pretraining stage. While tokenizing actions may introduce a marginal trade-off in low-level control precision compared to continuous heads, this

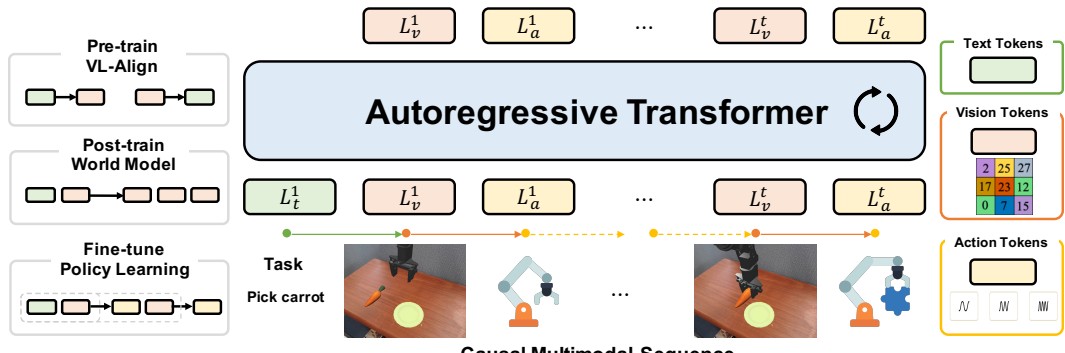

Figure 2: **Overview of the UniVLA framework.** Our model unifies information from different modalities into a discrete interleaved sequence, which is modeled using an autoregressive Transformer. To enable unified modeling, images are discretized using vector-quantized (VQ) encoders, while actions are transformed into the frequency domain and discretized via Discrete Cosine Transform (DCT) encoding. This causal multimodal sequence naturally preserves the temporal dynamics and causality required for real-world tasks. The model builds upon a pretrained vision-language model and follows a two-stage training strategy: (1) a post-training phase that adopts world-model training on large-scale datasets without requiring actions, and (2) a fine-tuning phase that interleaves actions into the sequence, enabling policy learning on downstream tasks.

unified architecture offers significant advantages in pretraining scalability and multimodal alignment. Furthermore, this foundation is inherently extensible, accommodating specialized action experts during fine-tuning for scenarios requiring high-precision actuation.

## 2.2 WORLD MODELS FOR ROBOTICS

World models Ha & Schmidhuber (2018); Hafner et al. (2019a); LeCun (2022) have gained widespread attention for their ability to capture and reason about the dynamics of the physical world. They have emerged as a cornerstone in a range of domains, including interactive video generation Bruce et al. (2024); Che et al. (2024), autonomous driving Hu et al. (2023a); Wang et al. (2024d;b); Gao et al. (2024), and robotics Du et al. (2023); Wu et al. (2023); Yang et al. (2023). Recent advances in robotics increasingly focus on general-purpose controllable video generation to simulate realistic and diverse robot-environment interactions. Visual Foresight Finn & Levine (2017) leverages action-conditioned video prediction with model-predictive control, enabling robots to plan manipulation tasks by forecasting future visual observations. UniSim Yang et al. (2023) builds a "universal simulator" trained on diverse visual datasets, capable of visualizing the effects of both high-level instructions (e.g., "open the drawer") and low-level controls in novel scenes. RoboDreamer Zhou et al. (2024) learns a compositional world model by factorizing video generation, facilitating the synthesis of novel action sequences. DREMA Barcellona et al. (2024) replicates scene dynamics and structure by integrating Gaussian Splatting with physics simulation. VLP Du et al. (2024) enables long-horizon visual planning by combining text-to-video generation with vision-language models as heuristic evaluators. DayDreamer Wu et al. (2023) extends Dreamer Hafner et al. (2019b) to real-world robotic platforms, while UVA Li et al. (2025b) proposes a joint video-action latent space to decouple video and action generation, achieving high accuracy and efficiency in policy inference. AdaWorld Gao et al. (2025) extracts latent actions from videos in a self-supervised manner and builds an autoregressive world model conditioned on these latent actions.

## 3 UNIFIED VISION-LANGUAGE-ACTION MODEL

In this section, we present the design of UniVLA, as illustrated in Figure 2. Unlike previous VLA models Kim et al. (2024); Black et al. (2024) that rely on ViT Dosovitskiy et al. (2021) for image encoding, our approach adopts an encoder-free architecture, converting all modalities into discrete tokens and learning them autoregressively. The overall design is simple yet effective, demonstrating strong scalability.

Our *unified paradigm* has two key aspects: first, it *unifies the learning of multiple modalities*, integrating various modality tokens into a shared representation space and employing a transformer for autoregressive learning; Second, it *unifies sequence modeling across tasks* through the natural interleaving of modalities, facilitating the seamless combination of tasks such as video generation, visual grounding, and action learning. In the following sections, we will introduce the method from the perspectives of *Unified Multimodal Model* and *Unified Multimodal Sequence Modeling*.

## 3.1 UNIFIED MULTIMODAL MODEL

As illustrated in Figure 2, our method unifies language, vision, and action modalities by converting each into discrete tokens and concatenating them into a single multimodal sequence $L$. Specifically, $L_t$, $L_v$, and $L_a$ denote the discrete token sequences for language, vision, and action, respectively, all drawn from a shared vocabulary. The superscript indicates the temporal step, with tokens interleaved across modalities to preserve temporal alignment.

For example, in the robotic manipulation task, a textual instruction is provided only at the beginning, followed by a naturally interleaved sequence of visual observations and actions. The language and vision tokenizers adopt the same design as Emu3 Wang et al. (2024c); visual observations are discretized using a VQ tokenizer Zheng et al. (2022), while actions are encoded using FAST Pertsch et al. (2025). To clearly demarcate modality boundaries, we employ special tokens—`boi` (begin of image), `eoi` (end of image), `boa` (begin of action), and `eoa` (end of action)—to encapsulate image and action tokens, respectively.

**Action Modeling**  We follow FAST Pertsch et al. (2025) and apply the Discrete Cosine Transform (DCT) to convert continuous action sequences into discrete action tokens. Specifically, given an action sequence within a time window, we define $L_a$ at a given time step as a sequence of action tokens $[T_1, \ldots, T_n]$. The raw action sequence $A_{1:H} = \{a_1, a_2, \ldots, a_H\}$ spans a window of size $H$, where each action $a_t$ is a $d$-dimensional vector. The FAST action tokenizer encodes $A_{1:H}$ into a discrete token sequence $[T_1, \ldots, T_n]$, with $n$ tokens drawn from a vocabulary of size $|V|$. Similar to natural language processing, action sequences can vary in token length, resulting in a variable-length ($n$) discrete representation.

**Training Objective**  Since all modality signals are transformed into discrete tokens, the training objective is simplified to a standard next-token prediction task using cross-entropy loss. To accommodate different task formats, we selectively include specific tokens in the loss computation, ensuring compatibility and flexibility across diverse tasks.

## 3.2 UNIFIED MULTIMODAL SEQUENCE MODELING

As shown in Figure 2, our multimodal sequence representation naturally captures the temporal dynamics and causal structure inherent in task execution. The embodied planning problem can be formulated as a Markov Decision Process (MDP), a general mathematical framework for decision-making in partially stochastic environments. For example, in the task of picking a carrot, the instruction and current observation inform the action; this action alters the environment, leading to a new observation that subsequently guides the next action. Building on this interleaved Markovian formulation, we unify a variety of tasks within a shared sequence modeling framework, and present the task-specific modeling strategies in the following.

**World Model (Post-train)**  Within the MDP framework, a world model aims to learn the dynamics of the environment by modeling the transition function $P(\mathbf{s}_{t+1}|\mathbf{s}_t, \mathbf{a}_t)$. The learned world model enables agents to simulate future trajectories, plan actions, and reason about consequences without direct interaction with the environment. Specifically, in the context of robotic tasks, we treat the language instructiom as a general form of action. Given the current observation $L_v^1$ and the instruction $L_t^1$, the world model need to predict future visual content. In this setting, we use the loss computed solely from the vision tokens as the supervisory signal, enabling the model to generate visual predictions conditioned on the given instruction and observed state. Sequence $S_v$ formulation is as follows:

$$S_v = \{L_t^1, L_v^1, L_v^2, ..., L_v^t\} \tag{1}$$

**Policy Learning (Fine-tune)**    Policy learning enables the agent to determine optimal actions based on both current observations and prior states, thereby effectively guiding task execution. In this setting, we employ a loss function computed solely from the action tokens. The sequence $S_a$ representing the interactions over time is formulated as:

$$S_a = \{L_t^1, L_v^1, L_a^1, L_v^2, L_a^2, \ldots, L_v^t, L_a^t\} \tag{2}$$

As illustrated in Figure 2, in this interleaved format, we adopt a two-stage training paradigm for robotic tasks. The model is initialized with a vision-language (VL) aligned checkpoint, endowing it with basic vision-language capabilities. The post-training stage leverages a world model objective to capture video dynamics, treating world modeling as a general visual learning task. Building upon the learned world model, the fine-tuning stage focuses on action learning to refine task-specific behaviors. We observe that incorporating the world model significantly enhances the efficiency and effectiveness of policy learning.

## 4 EXPERIMENTS

### 4.1 DATASET

**CALVIN.**    *CALVIN* Mees et al. (2022b) is a simulated benchmark tailored for evaluating long-horizon, language-conditioned robotic manipulation. It comprises four simulated environments (A, B, C, and D), each containing demonstration trajectories collected via human teleoperation. The benchmark encompasses 34 distinct manipulation tasks with a total of 1,000 unique language instructions. Performance is measured by the average number of successfully completed sub-tasks within a sequence. Standard evaluation protocols include the *ABC→D* and *ABCD→D* settings, which test a model's ability to generalize to unseen environments and compositions of long-horizon tasks.

**LIBERO.**    The *LIBERO* benchmark Liu et al. (2023) is a comprehensive suite for lifelong robotic manipulation, comprising four task suites with 10 tasks and 50 human demonstrations each. These suites are designed to evaluate different generalization abilities: *LIBERO-Spatial* tests spatial reasoning by varying layouts with fixed objects; *LIBERO-Object* assesses object-level generalization with varying objects in a fixed scene; *LIBERO-Goal* targets goal-conditioned behavior by varying task goals; and *LIBERO-Long* (*LIBERO-10*) features long-horizon, compositional tasks with diverse objects, layouts, and goals, challenging temporal and compositional reasoning.

**SimplerEnv.**    SimplerEnv Li et al. (2024d) serves as a simulation benchmark designed to evaluate the transferability and generalization capabilities of models trained on real-world video data. It incorporates diverse manipulation tasks on WidowX and Google Robot platforms with variations in lighting, object textures, colors, and camera viewpoints.

### 4.2 IMPLEMENTATION DETAILS

The model is a purely autoregressive Transformer with 8.5B parameters, identical to Emu3 Wang et al. (2024c). Images are tokenized via a VQ-based encoder with 8× spatial compression. Actions are encoded as frame-to-frame differences, normalized by the 1st/99th percentiles, and tokenized with FAST Pertsch et al. (2025), whose 1024-token vocabulary replaces the last 1024 IDs of the language tokenizer.

**Post-training Stage.**    In the post-training stage, we leverage large-scale robot-centric video datasets to study the effects of various post-training strategies on downstream policy learning. The model is initialized with pre-trained weights from the first stage of Emu3 Wang et al. (2024c). We curate a total of 622K videos from existing robotics datasets (details provided in the appendix), and identify the world model as the most effective post-training approach. During training, supervision is applied solely on the vision tokens. The model is trained for 30K steps with a batch size of 64.

**Fine-tuning Stage.**    During fine-tuning, the model is initialized with weights from the post-training stage and trained using a two-frame interleaved vision-action sequence with an action chunk size of 10. A cosine annealing learning rate schedule is applied, starting at $8 \times 10^{-5}$, and the loss is computed solely over action tokens. For the CALVIN benchmark, RGB observations from both third-person ($200 \times 200$) and wrist-view ($80 \times 80$) cameras are used. Training is conducted on A100

GPUs with a batch size of 192 for 8k steps. For the LIBERO benchmark, third-person and wrist-view RGB images (both at $200 \times 200$) are used to train a unified model with a batch size of 192 for 8k steps. A single model is evaluated across four task suites. For the SimplerEnv benchmark, single-view RGB observations are used with input resized to $256 \times 256$. Training is conducted on the Bridge-WidowX setup using a batch size of 128 for 20k steps, with an action chunk size of 5.

**Inference.** Our method adopts an interleaved vision–action training scheme, requiring supervision from future frames only during training. During inference, the model generates only action tokens without predicting future frames, conditioning on the current observed images. The action generation stops once the model predicts the `eoa` (end of action) token.

Additional implementation details on the post-training strategy, real-robot fine-tuning procedures, and autonomous driving experiments are provided in the appendix.

Table 1: **Long-horizon robotic manipulation evaluation on the CALVIN benchmark.** * denotes the use of video inputs.

| Method | Task | Tasks Completed in a Row | | | | | Avg. Len ↑ |
|---|---|---|---|---|---|---|---|
| | | 1 | 2 | 3 | 4 | 5 | |
| MCIL Lynch & Sermanet (2020) | ABCD→D | 0.373 | 0.027 | 0.002 | 0.000 | 0.000 | 0.40 |
| RT-1 Brohan et al. (2022) | ABCD→D | 0.844 | 0.617 | 0.438 | 0.323 | 0.227 | 2.45 |
| Robo-Flamingo Li et al. (2024c) | ABCD→D | 0.964 | 0.896 | 0.824 | 0.740 | 0.660 | 4.09 |
| GR-1* Wu et al. (2024) | ABCD→D | 0.949 | 0.896 | 0.844 | 0.789 | 0.731 | 4.21 |
| UP-VLA Zhang et al. (2025) | ABCD→D | 0.962 | 0.921 | 0.879 | 0.842 | 0.812 | 4.42 |
| RoboVLMs* Li et al. (2024b) | ABCD→D | 0.967 | 0.930 | 0.899 | 0.865 | 0.826 | 4.49 |
| **UniVLA*** | ABCD→D | **0.985** | **0.961** | **0.931** | **0.899** | **0.851** | **4.63** |
| MCIL Lynch & Sermanet (2020) | ABC→D | 0.304 | 0.013 | 0.002 | 0.000 | 0.000 | 0.31 |
| Robo-Flamingo Li et al. (2024c) | ABC→D | 0.824 | 0.619 | 0.466 | 0.331 | 0.235 | 2.47 |
| SuSIE Black et al. (2023) | ABC→D | 0.870 | 0.690 | 0.490 | 0.380 | 0.260 | 2.69 |
| GR-1* Wu et al. (2024) | ABC→D | 0.854 | 0.712 | 0.596 | 0.497 | 0.401 | 3.06 |
| MoDE* Reuss et al. (2024) | ABC→D | 0.962 | 0.889 | 0.811 | 0.718 | 0.635 | 4.01 |
| UP-VLA Zhang et al. (2025) | ABC→D | 0.928 | 0.865 | 0.815 | 0.769 | 0.699 | 4.08 |
| RoboVLMs* Li et al. (2024b) | ABC→D | 0.980 | 0.936 | 0.854 | 0.778 | 0.704 | 4.25 |
| Seer-Large Tian et al. (2024) | ABC→D | 0.963 | 0.916 | 0.861 | 0.803 | 0.740 | 4.28 |
| **UniVLA*** | ABC→D | **0.989** | **0.948** | **0.890** | **0.828** | **0.751** | **4.41** |

## 4.3 MAIN RESULTS

In this section, we evaluate our method on three simulation benchmarks: CALVIN (long-horizon tasks), LIBERO (diverse generalization), and SimplerEnv (real-to-sim manipulation). Our approach consistently achieves state-of-the-art performance across all settings.

**CALVIN Simulation Evaluation.** Table 1 presents the experimental results in the CALVIN benchmark. Our method achieves the highest performance on both the ABC→D and ABCD→D tasks, significantly outperforming previous approaches and demonstrating strong capabilities in multi-task learning and long-horizon planning.

**LIBERO Simulation Evaluation.** Following Zhao et al. (2025), we report the average success rate over 500 episodes for each task suite (Spatial, Object, Goal, Long). As shown in Table 2, UniVLA achieves the best overall performance across all LIBERO benchmark suites, with particularly significant gains on long-horizon tasks—raising the previous best from 69.0% to 94.0%. Compared to $\pi_0$ Pertsch et al. (2025), our method demonstrates superior performance on long-horizon tasks.

Table 2: **Comparison of different methods on the LIBERO benchmark.**

| Method | SPATIAL | OBJECT | GOAL | LONG | Average |
|---|---|---|---|---|---|
| DP* Chi et al. (2023) | 78.3% | 92.5% | 68.3% | 50.5% | 72.4% |
| Octo Team et al. (2024) | 78.9% | 85.7% | 84.6% | 51.1% | 75.1% |
| OpenVLA Kim et al. (2024) | 84.9% | 88.4% | 79.2% | 53.7% | 76.5% |
| SpatialVLA Qu et al. (2025) | 88.2% | 89.9% | 78.6% | 55.5% | 78.1% |
| CoT-VLA Zhao et al. (2025) | 87.5% | 91.6% | 87.6% | 69.0% | 81.1% |
| $\pi_0$-FAST Pertsch et al. (2025) | 96.4% | 96.8% | 88.6% | 60.2% | 85.5% |
| **UniVLA (single-frame)** | **97.0%** | **99.0%** | 92.6% | 90.8% | 94.8% |
| **UniVLA** | 95.4% | 98.8% | **93.6%** | **94.0%** | **95.5%** |

**SimplerEnv Simulation Evaluation.** Table 3 summarizes the performance across various manipulation policies on the Bridge-WidowX setup. Our approach demonstrates a significant improvement over prior methods, raising the average success rate from 42.7% to 69.8%. In particular, it shows marked improvements on previously difficult tasks, including stack block, put carrot and put spoon.

Table 3: **Evaluation on SimplerEnv-WidowX across various manipulation tasks.**

| Model | Put Spoon on Towel | | Put Carrot on Plate | | Stack Green on Yellow Block | | Put Eggplant in Yellow Basket | | Overall |
|---|---|---|---|---|---|---|---|---|---|
| | Grasp | Success | Grasp | Success | Grasp | Success | Grasp | Success | Success |
| RT-1-X Brohan et al. (2023) | 16.7% | 0.0% | 20.8% | 4.2% | 8.3% | 0.0% | 0.0% | 0.0% | 1.1% |
| Octo-Base Octo Model Team et al. (2024) | 34.7% | 12.5% | 52.8% | 8.3% | 31.9% | 0.0% | 66.7% | 43.1% | 16.0% |
| Octo-Small Octo Model Team et al. (2024) | 77.8% | 47.2% | 27.8% | 9.7% | 40.3% | 4.2% | 87.5% | 56.9% | 29.5% |
| OpenVLA Kim et al. (2024) | 4.1% | 0.0% | 33.3% | 0.0% | 12.5% | 0.0% | 8.3% | 4.1% | 1.0% |
| RoboVLMs Li et al. (2024b) | 70.8% | 45.8% | 33.3% | 20.8% | 54.2% | 4.2% | 91.7% | 79.2 | 37.5% |
| SpatialVLA Qu et al. (2025) | 20.8% | 16.7% | 29.2% | 25.0% | 62.5% | 29.2% | 100% | **100%** | 42.7% |
| CogACT Li et al. (2024a) | - | 71.1% | - | 50.8% | - | 15.0% | - | 67.5% | 51.3% |
| **UniVLA** | **83.3%** | **83.3%** | **74.0%** | **66.7%** | **95.8%** | **33.3%** | **100.0%** | 95.8% | **69.8%** |

## 4.4 In-Depth Analysis

This section presents an in-depth analysis of our unified framework, deriving critical design insights for future VLA models. We primarily focus on the **pivotal role of the world model**, illustrating how its integration as a post-training strategy fundamentally advances downstream policy learning capabilities (Table 4) and accelerates training efficiency (Table 5). We then turn to an ablation analysis of the module design. We find that even without post-training stage, incorporating visual prediction loss (Table 6a) and historical context (Table 6b) still contributes positively to policy learning.

**Effectiveness of World Model Post-Training.** Table 4 investigates the effects of different post-training strategies on downstream policy learning across various simulation benchmarks. Our results reveal that action-only post training suffers from limited transferability, primarily due to the non-uniformity of action spaces across diverse robot datasets—where variations in embodiments, control frequencies, and normalization result in heterogeneous distributions. In contrast, most vision-based post-training approaches significantly enhance policy learning and demonstrate superior generalization without relying on explicit action labels, thereby highlighting the crucial role of visual learning in transferability. Among these, the world model post-training approach yields the most substantial gains, enhancing both generalization and long-horizon planning capabilities. A comparison with text-to-image (T2I) training emphasizes the importance of modeling temporal dynamics in video data, while video-only training reveals the importance of textual guidance for state transitions. Notably, this world model requires no action annotations, enabling scalable learning from large-scale video data and offering a promising direction for future VLA research.

**Data and Training Efficiency for Fine-tuning.** Table 5 shows that post-training substantially enhances downstream policy learning efficiency. On the CALVIN benchmark (Table 5a), our method achieves higher success rates using only 10% of the fine-tuning data, outperforming prior approaches such as GR-1 Wu et al. (2024) and RoboVLMs Li et al. (2024b). In addition, Table 5b highlights improved training efficiency, as the model rapidly converges with fewer fine-tuning iterations. The Simpler-Env results further demonstrate the effectiveness of world-model-based post-training for efficient policy adaptation across diverse robotic setups. While similar effects are observed in latent-action methods Ye et al. (2025); Chen et al. (2024b); Gao et al. (2025), our world model offers a simpler paradigm without latent actions, achieving better transferability.

**Effectiveness of Visual Prediction.** While post-training proves effective, it is also crucial that the model demonstrates strong performance without relying on it. As shown in Table 6a, our findings

Table 4: **Effectiveness of World Model Post-Training.** We compare different post-training strategies by fine-tuning only with action prediction on the downstream benchmarks.

| Post-training Stage | | | Generalization | | Long-horizon | |
|---|---|---|---|---|---|---|
| Strategy | Sequence | Supervision | LIBERO | SimplerEnv-WidowX | LIBERO-Long | CALVIN |
| | | | 48.5 | 0.0 | 17.4 | 1.46 |
| action prediction | $T, I, A$ | action | 43.9 (-4.6) | 0.0 | 10.6 (-6.8) | 0.52(-0.94) |
| text-to-image | $T, I$ | vision | 69.8 (+21.3) | 6.3 (+6.3) | 55.8 (+38.4) | 3.79 (+2.33) |
| video prediction | $I_1, ..., I_t$ | vision | 78.9 (+30.4) | 17.7 (+17.7) | 80.8 (+63.4) | 3.59 (+2.13) |
| world model | $T, I_1, ..., I_t$ | vision | **94.2** (+45.7) | **64.6** (+64.6) | **89.2** (+71.8) | **4.61** (+3.15) |

Table 5: **Post-training enables data-efficient and training-efficient downstream policy learning.**

(a) **Data efficiency comparison.**

| Method | Data | CALVIN |
|---|---|---|
| RT-1 Brohan et al. (2022) | 10% | 0.34 |
| MT-R3M Nair et al. (2022) | 10% | 0.61 |
| HULC Mees et al. (2022a) | 10% | 1.11 |
| GR-1 Wu et al. (2024) | 10% | 2.00 |
| RoboVLMS Li et al. (2024b) | 10% | 2.52 |
| **UniVLA** (w/o post-train) | 10% | 0.15 |
| **UniVLA** | 10% | **3.19** |

(b) **Training efficiency comparison.**

| Fast convergence (CALVIN) | | | |
|---|---|---|---|
| Training Iters | 2k | 4k | 8k |
| w/o post-train | 0.37 | 0.82 | 1.46 |
| w/ post-train | 4.21 | 4.56 | 4.61 |
| **Fast adaptation (SimplerEnv-Bridge)** | | | |
| Method | Batch size | Iters | Success |
| RoboVLMs Li et al. (2024b) | 128 | 50k | 37.5 |
| **UniVLA** | 128 | 12k | 64.6 |

Table 6: **Ablation study on the visual prediction and historical context in policy learning.**

(a) **Effectiveness of visual prediction.**

| Post-train | Visual prediction | CALVIN | LIBERO |
|---|---|---|---|
| ✓ | | 4.61 | 94.2 |
| | ✓ | 4.42 | 88.7 |
| | | 1.46 | 48.5 |

(b) **Effectiveness of history context.**

| Observations | | Avg. Len ↑ |
|---|---|---|
| History Window | Current + History | |
| 0 | 1 + 0 | 4.26 |
| 10 | 1 + 1 | 4.61 |
| 10 | 1 + 2 | 4.43 |
| 20 | 1 + 2 | 4.47 |

indicate that, even without post-training, fine-tuning with visual loss supervision—leveraging the autoregressive nature of the model—naturally integrates world model learning into the policy learning process. This approach leads to a significant improvement in the model's performance.

**Effectiveness of History Context.** History context—comprising past observations and actions—provides valuable guidance for robot planning. In this section, we investigate the appropriate length of the history window during the fine-tuning stage. As shown in Table 6b, our ablation study on the CALVIN benchmark examines the impact of varying history window lengths. Incorporating a history window significantly improves performance (from 4.26 to 4.61). However, extending the window beyond a certain length yields diminishing returns, suggesting that recent observations carry the most predictive value, consistent with the Markov property in sequential planning.

## 4.5 BROADER APPLICATIONS

**Real-world ALOHA Tasks.** We further validate the effectiveness of our approach on the real-world ALOHA platform through two tasks: *pouring water* and *folding clothes*. Our model relies exclusively on visual observations, without access to any state information.

As shown in Table 7, we decompose the pouring task into three sequential subtasks: *right-hand grasping the bottle*, *left-hand grasping the cup*, and *pouring*. UniVLA demonstrates competitive performance against $\pi_0$ Black et al. (2024) in real-world ALOHA tasks. While $\pi_0$ achieves a marginally higher success rate in the grasping stage, both methods exhibit comparable performance in the pouring stage. All results are averaged over 8 trials. Notably, we observe that precise pouring remains a challenge for both methods. Interestingly, we observe that world model post-training yields a substantial performance improvement even in real-world experiments. Additional details and demonstration videos are provided in the appendix.

Table 7: **Comparison across real-world ALOHA manipulation tasks.**

| Method | Right-hand Grasp Bottle | Left-hand Grasp Cup | Pour Water | Overall |
|---|---|---|---|---|
| $\pi_0$ Black et al. (2024) | 87.5 | 75.0 | 37.5 | 37.5 |
| **UniVLA** (w/o world model) | 12.5 | 0.0 | 0.0 | 0.0 |
| **UniVLA** (w/ world model) | 87.5 | 62.5 | 37.5 | 37.5 |

As shown in Figure 3, despite the precision loss from discrete tokens, our method is still able to perform fine-grained tasks such as folding clothes.

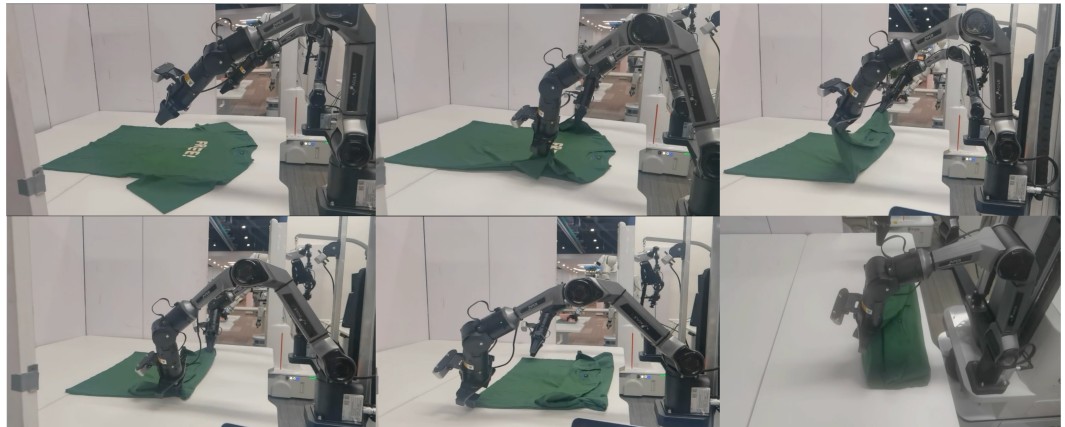

Figure 3: **Visualizations of the *folding clothes* task performed by our method.**

**End-to-end Learning for Autonomous Driving.** To further explore the potential of our method, we perform a preliminary transfer to the autonomous driving domain by finetuning the model on the NAVSIM benchmark. Notably, our method is a pure autoregressive, token-based framework, modeling the driving task as causal sequence prediction over discretized multimodal tokens. Despite using only front-view camera inputs—without relying on BEV representations or multi-sensor fusion—our model achieves powerful performance on the NAVSIM test set. Notably, world model post-training is also effective in the driving domain and can lead to significant performance improvements. These results highlight the strong potential of our method for broader real-world applications.

Table 8: **Broader applications of UniVLA for end-to-end autonomous driving on the NAVSIM.** MC: Multi Camera. L: LiDAR. FC: Front Camera.

| Method | Model | Input | NC↑ | DAC↑ | EP↑ | TTC↑ | C↑ | PDMS↑ |
|---|---|---|---|---|---|---|---|---|
| Human | – | – | 100.0 | 100.0 | 87.5 | 100.0 | 99.9 | 94.8 |
| Ego Status MLP | – | Ego State | 93.0 | 77.3 | 62.8 | 83.6 | 100.0 | 65.6 |
| VADv2 Chen et al. (2024a) | BEV-based | MC | 97.9 | 91.7 | 77.6 | 92.9 | 100.0 | 83.0 |
| UniAD Hu et al. (2023b) | BEV-based | MC | 97.8 | 91.9 | 78.8 | 92.9 | 100.0 | 83.4 |
| Transfuser Chitta et al. (2022) | BEV-based | MC&L | 97.7 | 92.8 | 79.2 | 92.8 | 100.0 | 84.0 |
| **UniVLA** | Auto-regressive | FC | 96.9 | 91.1 | 76.8 | 91.7 | 96.7 | 81.7 |
| **UniVLA (w/ world model post-train)** | Auto-regressive | FC | 98.3 | 93.8 | 80.0 | 94.2 | 100.0 | 85.6 |

## 5  CONCLUSION

In this paper, we present UniVLA, a unified framework for vision–language–action modeling that bridges heterogeneous modalities through a shared token space and models them autoregressively. The proposed unified design facilitates deeper cross-modal integration and inherently supports flexible multimodal tasks. By leveraging a world model trained to capture dynamics and causality from videos, we observe significant improvements in downstream policy learning, both in terms of performance and efficiency. Extensive simulation experiments further demonstrate the model's strong generalization ability, efficient policy learning, and broad applicability across diverse domains. These findings highlight the great potential of our method as a new paradigm for vision–language–action modeling.

**Limitations and Future Work** Our exploration of post-training scalability is currently limited by compute availability, yet early indicators show promise for scaling to extensive video datasets. Although our unified framework excels in cross-modal learning, further research is needed to seamlessly integrate reinforcement learning for robust policy adaptation. Regarding real-world deployment, we have validated the feasibility of our approach on tasks such as dual-arm pouring and cloth folding; however, generalizing to more complex, open-ended manipulation tasks remains an open challenge.

ACKNOWLEDGMENTS

We would like to thank BAAI for providing the data and computing resources. This work was supported in part by the National Key R&D Program of China (No. 2022ZD0116500), the National Natural Science Foundation of China (No. U21B2042, No. 62320106010).

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
