APPENDIX

## A  IMPLEMENTATION DETAILS

**Post-training Stage**  We began by selecting several high-quality robotics datasets for post-training, as summarized in Table 9. To account for differences in data collection frequencies across datasets, we applied dataset-specific frame sampling intervals to ensure that the temporal gap between keyframes is approximately one second. We further filtered out video sequences containing fewer than six frames, as well as those lacking corresponding text instructions. Due to the large number of videos from the Kuka Kalashnikov et al. (2018) dataset, we randomly retained 100k videos to prevent it from dominating the overall training data.

Table 9: **Post-training datasets**.

| Dataset | Source | Data Type | Raw Videos | Used Videos | Interval |
|---|---|---|---|---|---|
| RT-1 Brohan et al. (2022) | Real | Text, Video, Action | 87212 | 84084 | 3 |
| BridgeV2 Walke et al. (2023) | Real | Text, Video, Action | 60064 | 28083 | 5 |
| DROID Khazatsky et al. (2024) | Real | Text, Video, Action | 275997 | 145641 | 15 |
| Kuka Kalashnikov et al. (2018) | Real | Text, Video, Action | 580392 | 100000 | 3 |
| TOTO Zhou et al. (2023) | Real | Text, Video, Action | 902 | 899 | 20 |
| Taco Play Rosete-Beas et al. (2023) | Real | Text, Video, Action | 3242 | 3242 | 5 |
| FMB Luo et al. (2023) | Real | Text, Video, Action | 8611 | 7876 | 5 |
| Berkeley autolab ur5 Chen et al. | Real | Text, Video, Action | 896 | 896 | 5 |
| VIOLA Zhu et al. (2023) | Real | Text, Video, Action | 135 | 135 | 15 |
| Cmu Play Fusion Chen et al. (2023) | Real | Text, Video, Action | 576 | 576 | 10 |
| Utaustin Mutex Shah et al. (2023) | Real | Text, Video, Action | 1500 | 1500 | 10 |
| CALVIN Mees et al. (2022b) | Sim | Text, Video, Action | 22966 | 22966 | 5 |
| LIBERO Liu et al. (2023) | Sim | Text, Video, Action | 3386 | 3386 | 10 |
| ManiSkill2 Gu et al. (2023) | Sim | Text, Video, Action | 30213 | 193273 | 10 |
| SSV2 Goyal et al. (2017) | Real | Text, Video | 220847 | 220847 | 1 |

For the experiments in Table 4, to ensure a fair comparison of different post-training strategies, all models are trained on the same dataset (excluding SSV2 Goyal et al. (2017), which does not contain action annotations), with only the post-training strategy varied. For the *action prediction* task, we organize the input as $(T, I, A)$, where $T$ denotes the text instruction, $I$ the image observations, and $A$ the action sequence. During training, only the action tokens $A$ are supervised in the loss computation. For the *text-to-image* task, the input is organized as $(T, I)$, where $T$ denotes the input text and $I$ denotes the target image. During training, the loss is only computed on the vision tokens corresponding to $I$. For the *video prediction* task, the input is organized as $(I_1, ..., I_t)$, where $I$ denotes the video frame. During training, the loss is computed on the vision tokens. For the *world model* task, the input is organized as $(T, I_1, ..., I_t)$, where $T$ denotes the input text, $I$ denotes the video frame. During training, the loss is computed on the vision tokens.

During training, the observations are resized to 256×256, using six frames as input, with the maximum sequence length set to 6400. We perform full-parameter training for 50k steps using 32 A100 GPUs (40GB), which takes approximately 4–5 days.

**Simulation Finetuning**  The training setup is described in the main paper. We adopt full-parameter training, and for evaluation, we follow the testing protocols of OpenVLA Kim et al. (2024) and RoboVLMs Li et al. (2024b) across various benchmarks. By default, our model is trained using video-format sequences; however, it also supports fine-tuning with image-format sequences. In the ablation study evaluating the effect of visual prediction, when post-training is not applied, the visual token weight is set to 0.5 while the action token weight is set to 1.0, in order to maintain balance between the two modalities.

**Real-robot Finetuning**  For real-world evaluation, we conduct experiments on the ALOHA platform, using images captured from three perspectives: `cam high`, `wrist left`, and `wrist right`. The real-robot is controlled using end-effector (EE) pose. All input images are resized to a

resolution of 128×128. The model outputs a 14-dimensional action vector. The action chunk size is set to 20. For each task, we train for 8k steps with a batch size of 256. The learning rate is set to $5 \times 10^{-5}$, and all other settings remain consistent with the above. We also leverage world model pretraining, using video-based post-training on a collected real-aloha dataset (Table 10). Interestingly, this post-training provides substantial benefits even when transferring to real-robot execution.

# B  REAL-ROBOT EXPERIMENTS

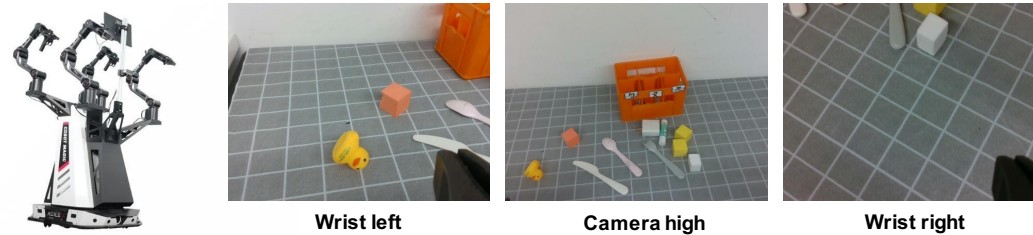

**Wrist left**  **Camera high**  **Wrist right**

Figure 4: **Real-world setup of the AgileX Cobot Magic dual-arm robot.** The system is equipped with three Orbbec RGB cameras for visual observation: one mounted on the left wrist, one on the right wrist, and one positioned above for a high-angle view.

## B.1  ALOHA EXPERIMENTAL SETUP

The robotic platform used in this paper is **AgileX Cobot Magic V2.0**, a dual-arm robot. As shown in Figure 4, the robot is equipped with two arms and three camera views, enabling it to perform a variety of manipulation tasks. For example, Figure 5 illustrates a range of manipulation tasks collected from real-world scenarios.

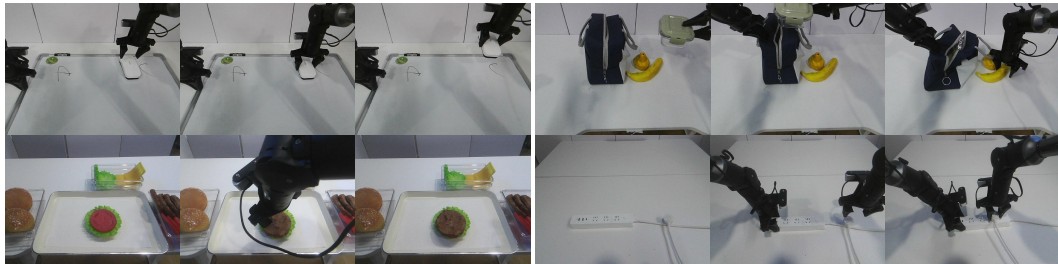

Figure 5: **Real-world task examples.** These include diverse tasks such as wiping a whiteboard, organizing tableware, making a burger, and plugging in a connector.

**Real-World Task Collection**  Table 10 provides a summary of the real-world data collected from the physical robot, recorded at an actual frequency of 30 Hz. A total of 8 tasks were included, with each task collecting approximately 500 trajectories on average. During preprocessing, static frames at the beginning and end of each trajectory were filtered out.

Table 10: **Real-world task trajectories.**

| Fold Clothes | Clear Desk | Store Glasses | Food Packing | Pour Water | Clean Blackboard | Insert Plug | Make Hamburger |
|:---:|:---:|:---:|:---:|:---:|:---:|:---:|:---:|
| 528 | 500 | 500 | 500 | 496 | 500 | 500 | 640 |

**Data Processing**  To reduce redundancy and improve training efficiency, we select keyframes based on thresholding the changes in recorded action joint values. For each selected sequence, the action chunk is normalized by subtracting the joint values of the first frame.

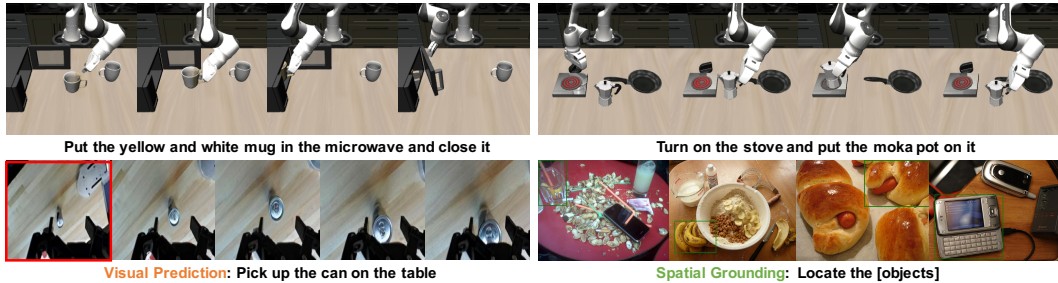

**Figure 6: Multimodal capabilities of UniVLA**. Top: Action outputs for executing long-horizon tasks in the LIBERO benchmark. Bottom: Visual predictions and spatial grounding demonstrating the model's spatiotemporal understanding. The red box marks the current observation; green boxes indicate predicted object detections.

## B.2 ALOHA EXPERIMENTS

For real-world experiments, we perform world model post-training using video data collected from the ALOHA platform, consisting of about 1,000 videos (Pouring water and fold clothes).

**Real-world Latency**   For real-world experiments, model inference is conducted remotely via network communication on an NVIDIA A100 GPU (40GB). The dual-arm robotic platform (AgileX Cobot Magic) receives three image observations, each at a resolution of 128×128 pixels. As shown in Table 11, actions are predicted in discrete chunks of 20 steps, corresponding to approximately a 3.3-second motion window. Each inference step on the model requires roughly 2.1 seconds, and when accounting for communication and data I/O overhead, the total system latency amounts to approximately 3.2 seconds. Consequently, the system generates 20 action steps every 3.2 seconds. These predicted actions are then executed sequentially over the 20 timesteps, with each step interpolated five times to achieve smoother and more precise control.

Table 11: **Inference latency in real-world deployment.**

| Setting | Action Chunk Size | Inference Time | Total Latency | Action Steps/sec |
|---|---|---|---|---|
| Real ALOHA | 20 steps | 2.1 s | 3.2 s | 6.3 |
| Real ALOHA, w/ vllm | 20 steps | 1.5 s | 2.6 s | 7.7 |

## C  AUTONOMOUS DRIVING EXPERIMENTS

**NAVSIM Setup**   The NAVSIM dataset Dauner et al. (2024), resampled from OpenScene to emphasize challenging scenarios, is currently one of the most established end-to-end evaluation benchmarks in the autonomous driving domain. The dataset is divided into two parts: Navtrain and Navtest, comprising 1,192 scenarios for training and validation, and 136 scenarios for testing.

For model training, the input images are resized to a resolution of 512×288. We follow the standard training setup, using the current image frame and ego status to predict trajectories for the next 8 frames. Both the action and ego status are encoded using the fast tokenizer. For post-training of the world model, we leverage six consecutive vision–action pairs, jointly supervising both vision and action outputs.

## D  MULTIMODAL CAPABILITY

As illustrated in Figure 6, we qualitatively showcase the model's ability to interleave multiple modalities—action, language, and vision—within a unified framework. This design enables policy learning for embodied control, spatial reasoning through language output, and future state prediction via visual output, highlighting the model's capacity for generalizable multimodal understanding.

# E    USE OF LLMS

Large Language Models (LLMs) are used for polishing writing in this manuscript.