# OpenReview forum: "Unified Vision-Language-Action Model"
_ICLR.cc/2026/Conference — ICLR 2026 Poster_

### Official Review · Reviewer_6H2r · 2025-10-26

**Soundness:** 4
**Presentation:** 4
**Contribution:** 3
**Rating:** 8
**Confidence:** 4

**Summary:**

The paper “UniVLA: Unified Vision-Language-Action Model” introduces a fully tokenized, autoregressive architecture that jointly models vision, language, and action within a single transformer-based sequence framework. Unlike conventional VLA systems that fuse pretrained visual encoders with language-conditioned action heads, UniVLA treats all three modalities as discrete tokens within a shared vocabulary, enabling unified cross-modal reasoning and flexible task conditioning.
The framework operates in two stages: (1) Post-training (world modeling): learning causal video dynamics and temporal prediction from large-scale robot-centric videos without action supervision. (2) Fine-tuning (policy learning): conditioning on interleaved visual and textual tokens to predict action tokens.

The approach achieves state-of-the-art results across CALVIN (4.63 average length vs. 4.49 for RoboVLMs), LIBERO (95.5% vs. 85.5% for π₀-FAST), and SimplerEnv (69.8% vs. 42.7%). It also transfers to real-world ALOHA robotic tasks and autonomous driving (NAVSIM), showing versatility of the unified token-based modeling.

**Strengths:**

- UniVLA’s key novelty lies in representing vision, language, and action as discrete tokens under a shared autoregressive framework. This eliminates separate encoders and heads, resulting in a conceptually elegant, flexible design. The framework’s success on both robotics and driving underscores its generality.

- The paper demonstrates that pretraining on large-scale video prediction (without action labels) significantly enhances downstream policy efficiency and long-horizon reasoning. Table 4’s analysis is particularly convincing, showing +45–70% relative improvement over action-only post-training.

- UniVLA consistently outperforms prior VLA baselines (π₀, FAST, OpenVLA, GR-1, RoboVLMs) across three simulation suites and two real-world domains. The inclusion of both robotic manipulation and driving tasks makes the evaluation unusually broad for a single model.

- The paper systematically analyzes data efficiency (Table 5a), training efficiency (Table 5b), world model variants, and historical context. The insights—for instance, diminishing returns beyond 2-step history—are practical for future VLA design.

- Demonstrating performance gains on the physical ALOHA robot (pouring, folding) and quantifying inference latency (≈3s per 20-step chunk) is commendable. Few VLA papers provide such detailed real-world evaluation.

**Weaknesses:**

- The model borrows heavily from existing architectures (Emu3 for transformer backbone, FAST for action tokenization, VQ-VAE for image tokens) and primarily unifies them under one autoregressive umbrella. While impactful, the approach feels more like a pragmatic consolidation than a paradigm shift.

- The paper includes numerous benchmarks, ablations, and cross-domain studies—sometimes at the expense of clarity. A concise ablation visualization or conceptual diagram summarizing findings could make the narrative more digestible.

- The authors claim that token unification enables stronger “causal reasoning,” but this is not analytically substantiated. There’s little formal justification for why autoregressive modeling over discrete tokens inherently captures causal structure better than multimodal fusion.

- The method’s scalability depends on massive video post-training (622k videos, 32×A100 GPUs for 5 days). The claim of “data efficiency” holds mainly at fine-tuning, not at pretraining. Real-world feasibility for smaller labs remains questionable.

- Although the paper reports strong quantitative results, it does not deeply analyze potential loss of control granularity introduced by discrete action tokenization, especially for fine-grained manipulation tasks (e.g., folding clothes).

**Questions:**

- How sensitive is UniVLA to the specific choice of discretization (VQ encoder, DCT-based FAST)? Could learned continuous latent codes outperform these fixed tokenizers?

- During post-training, how much of the improvement stems from video scale vs. the world model objective itself?

- Can UniVLA generalize to reinforcement learning settings where rewards guide action prediction rather than imitation?

- How does token-level autoregression scale with longer action horizons (e.g., >1,000 tokens per rollout)?

- Is there any observed degradation when transferring from simulated to real-world data, especially under domain shifts in lighting and textures?

---

> ### Author Response · Authors · 2025-11-20
> **PART1**
>
> We sincerely thank the reviewer for the insightful comments and constructive suggestions. We address each point in detail below.
>
> **W1: Paradigm novelty.**
>
> Thank you for the insightful comment. We agree that the primary contribution of our work does not lie in architectural novelty, but rather in the **task formulation and modeling paradigm**. Prior approaches predominantly rely on image-level VLMs and perform policy learning at the image level. In contrast, our work reframes the problem from a **video-centric perspective**, aiming to train a video foundation model that learns policies directly from video sequences—by placing the **causal task trajectories (T→I→A→I→A)** into the model in chronological order. Moreover, we introduce **visual supervision**, showing that it plays a crucial role in strengthening VLA learning and significantly improves generalization. Our experiments further demonstrate that policy learning transferred from such a video foundation model provides clear benefits in both performance and training efficiency.
>
> Furthermore, we emphasize the **simplicity and scalability of this framework**: all modalities are represented as discrete tokens, and a single autoregressive loss is used to unify modeling across vision, language, and action.
> This design intentionally avoids architectural complexity, favoring a clean and scalable formulation that delivers broad applicability and strong empirical performance.
>
> **W2: Concise ablation visualization.**
>
> Thank you for the suggestion. The core insight we aim to highlight in this paper is indeed captured in **Table 4**, which demonstrates the **substantial gains brought by world-model post-training for VLA polcy learning**. The subsequent ablations further reinforce this finding from the perspectives of **data efficiency** and **training efficiency**. Following your recommendation, we will add a concise summary of the key insights at the beginning of the experiments section to present the main contributions more clearly and improve overall readability.
>
>
> **W3: About causal reasoning.**
>
> We would like to clarify that the causal properties of UniVLA do not stem from the use of discrete autoregressive tokens alone. Instead, as illustrated in **Fig. 2**, **our modeling formulation preserves the full causal structure of a trajectory—i.e., [instruction, observation₁, action₁, observation₂, action₂, …]**—and feeds it into the model in a temporally ordered manner. The **causal autoregressive training objective** then enables the model to explicitly learn these underlying dependencies.
>
> In contrast to prior approaches that typically predict actions from a single observation (i.e., a momentary, frame-level prediction), our formulation **learns over entire trajectories**, capturing how observations and actions evolve over time. This trajectory-level modeling leads to much richer causal understanding.
>
> Consistent with this, when we perform inference without updating the observations and allow the model to freely roll out the trajectory, visualizations show that its autoregressive predictions of future observations and actions remain highly consistent, highlighting its ability to exploit learned causal dependencies.
>
> **W4: Data efficiency for finetuning.**
>
> We will clarify in the paper that the reported **data efficiency specifically refers to fine-tuning**. Analogous to the classic pretraining and fine-tuning paradigm in large language models, our intention is to highlight the value of a **video foundation model for VLA models**. Our ambition is that, in the future, a well-trained video foundation model will enable smaller labs to achieve strong downstream performance primarily via fine-tuning, without the need for extensive data collection or training from scratch.
>
> **W5: Potential loss of discrete action tokenization.**
>
> While discretization inevitably introduces some information loss, the impact in our setting is minimal. For example, after normalization and tokenization, the precision error of the ground-truth trajectory is approximately 0.005. Importantly, FAST’s DCT-based encoding allows **adjustable precision via the scale parameter**, meaning that quantization errors can be further mitigated simply by increasing the number of tokens when needed.
>
> In practice, this discretization error does not noticeably affect control performance. During real-world execution (e.g., folding clothes), each predicted action is additionally interpolated 5× per step to generate smooth control signals. Furthermore, real robots inherently exhibit execution noise and actuation inaccuracies, making the small quantization error introduced by FAST insignificant by comparison.
>
> Therefore, although discretization may introduce theoretical information loss, our real-world experiments consistently show that its practical impact on control performance is minimal.

---

> ### Author Response · Authors · 2025-11-20
> **PART2**
>
> **Q1: Choice of discretization.**
>
> The choice of discretization was motivated by our assumption that discrete autoregressive modeling converges faster. The VQ encoder and its configuration are currently tied to EMU3, and we expect that future, more efficient compression methods—such as a 16× (currently 8×) tokenizer—will further improve performance.
>
> | Action Modeling Method | calvin ABCD |
> |-----------------------|------------|
> | OpenVLA bin-based     |   3.98     |
> | FAST DCT-based            |   4.26     |
> | Flow matching continous | 3.36  |
>
> Regarding action modeling strategies, as shown in the table, we compared the original bin-based modeling in OpenVLA with the FAST-based approach. Across both performance and encoding efficiency, FAST demonstrates clear advantages. Furthermore, we experimented with flow-matching for action modeling, but found that both its performance and training efficiency are inferior to the FAST approach within our framework. Current experiments are conducted on the single-frame setting of the CALVIN benchmark, with $8,000$ training steps.
>
> **Q2: About post-training.**
>
> This issue can be partially addressed by the comparisons in **Table 4**. When comparing training with the world model objective versus alternatives such as video-based training, the improvement achieved by the world model is significant, demonstrating the effectiveness of the world model objective. Furthermore, we have supplemented additional experiments on data scale, as shown in the table below.
>
> | Data Scale  | Calvin ABCD | simpler-env |
> |------------|-------|-------|
> | ~100k       | 4.42 | 57.4 |
> | 622k       | 4.61 | 69.8 |
>
> Moreover, as shown in **Appendix Table 10**, the benefits of the world model objective are also clearly observed on real-world deployment.
>
> In fact, 622k videos do not constitute a particularly large scale. We expect that scaling pretraining to larger video datasets will further strengthen the model’s capabilities. However, given the current saturation of simulation benchmarks, this improvement is likely to be observed primarily on real-world deployments.
>
> **Q3: Generalize to reinforcement learning setting.**
>
> UniVLA can indeed be **naturally adapted to reinforcement learning (RL) settings**. It **outputs multimodal discrete tokens**, which aligns well with token-level RL commonly used in LLMs and also enables multimodal RL. By rolling out different action tokens, UniVLA can explore diverse trajectories and receive rewards for RL optimization.
>
> We think **the core challenge lies in designing effective reward functions**. A simple approach is to use consistency with expert trajectories as the reward, but this closely resembles imitation learning. Future work should explore how to define more reasonable rewards and how to fully leverage the advantages of our unified multimodal representation.
>
> **Q4: Longer action horizons.**
>
> We would like to clarify that **the number of action tokens is primarily determined by the size of the action window**, i.e., how many future steps the model predicts at each inference. The token count can vary significantly between simulation and real-world deployment.
>
> In simulation, we typically use a chunk size of 10 with an action dimensionality of 7, resulting in an average of 20-30 tokens after FAST encoding. On the real-world ALOHA, we use a chunk size of 20 with an action dimensionality of 14, yielding an average of 100–200 tokens.
>
> Regarding the suggestion of using 1000 action tokens, this is not very realistic. We have experimented with longer action windows—extending from 10→20 in simulation and 20→30 on the real robot. For the real robot, a chunk size of 30 would correspond to at most ~300 action tokens. As expected, performance inevitably degrades due to the model needing to perform long open-loop predictions without intermediate observations. This degradation is natural and reflects the accumulation of small prediction errors over extended horizons.
>
> In summary, our current choice of action window and resulting token count represents a practical trade-off between prediction horizon and stable performance.
>
> **Q5: Robustness under real-world experiments.**
>
> In practice, we find that **data augmentation plays an even more critical role in real-robot training**. Unlike simulation, where training and testing data are highly consistent, real-robot scenarios exhibit greater discrepancies between training and testing, along with additional noise. Specifically, we apply lighting, color, and random-shift augmentations, which substantially improve robustness to variations in illumination and texture. In real-robot evaluations, we tested scenarios with varying lighting conditions and noisy backgrounds, and observed that without these augmentations, the model’s performance becomes noticeably less stable.

---

### Official Review · Reviewer_Wn1i · 2025-10-31

**Soundness:** 4
**Presentation:** 3
**Contribution:** 3
**Rating:** 6
**Confidence:** 3

**Summary:**

This paper introduces a novel architecture, UniVLA, which encodes three modalities—image, text, and action—into discrete tokens. This enables joint modeling in an autoregressive manner, achieving better cross-modal alignment. Based on this multi-modal discrete token autoregressive paradigm, UniVLA can unify the world model training and visuomotor policy within a single model. It also demonstrates that world model training improves policy learning, especially in long-horizon tasks and OOD scenarios. The authors achieved promising results on classic VLA benchmarks, including CALVIN, LIBERO, and Simpler-Env, demonstrating the effectiveness of their method.

**Strengths:**

- The UniVLA architecture is meaningful: it discretizes images, text, and actions into tokens and trains them under a unified autoregressive paradigm. This enables stronger cross-modal alignment via diverse proxy tasks, such as world-model training, policy learning, and multimodal understanding.
- The authors demonstrate that world-model training is an effective auxiliary task that benefits downstream policy learning.
- UniVLA achieves competitive results on several standard benchmarks, showing improvements over some classic VLA models.

**Weaknesses:**

- Concern about inference speed. The authors use Emu3-8.5B as the autoregressive base, which is non-trivial in size for a VLA backbone. Moreover, action prediction requires generating a sequence of tokens, which inevitably slows inference. This is especially problematic in real-robot experiments, where limited on-device compute leads to a low action-prediction rate. The authors should further discuss UniVLA's deployment in simulation and on real hardware, its baseline compute requirements, and the concrete inference frequency.
- Limitations of real-world experiments. As a VLA model, its performance in the real world is a crucial metric of its quality. However, the experimental section only outlines the basic setup of the real-robot evaluation in L454–457, without reporting UniVLA’s success rate on real tasks or comparing its real-world performance against other competitive VLA models. This omission hinders a fuller assessment of UniVLA’s contributions.
- The authors should discuss how much of the current performance gain stems from Emu3's inherent multimodal discrete-token modeling. While we acknowledge that modeling vision, language, and action uniformly as discrete tokens is a reasonable approach, would the performance remain comparable if this paradigm were transplanted to other backbones? Notably, OpenVLA and Pi-FAST were not pretrained with such multimodal discrete-tokenization prediction.

**Questions:**

- Have the authors analyzed and compared different action encoding strategies? Within the autoregressive action-prediction paradigm, there are many ways to encode discrete tokens (e.g., FAST, VQ-VAE), and one could also adopt continuous latent encodings.
- Did the authors incorporate multimodal understanding tasks within this multimodal discrete-token modeling paradigm? In the leftmost panel of Figure 1, we notice an illustration of a multimodal grounding task, yet Section 3.2 does not describe the corresponding training procedure.
- Further discussion on architecture. Modeling discrete tokens across the three modalities is meaningful for aligning them, but it may be insufficient for VLA. For example, in PI-0.5, the first training stage jointly models vision, language, and action (with images not discretized and actions discretized via FAST), yet the second stage still introduces an action expert to produce actions, ensuring precise action modeling. Have the authors examined the difference between predicting actions solely with a VLM and adopting a dual-system–style architecture?

---

> ### Author Response · Authors · 2025-11-20
> **PART1**
>
> We sincerely thank the reviewer for the insightful comments and constructive suggestions. We address each point in detail below.
>
> **W1: Inference speed.**
>
> We thank the reviewer for raising the concern regarding inference speed. Many prior methods, such as Open-VLA and Fast, perform sequential action prediction, which can be accelerated using techniques developed for large models, e.g., KV-cache and vLLM. In contrast, thanks to our interleaved image-action training paradigm, UniVLA conditions on real observations and only **predicts action chunks** (e.g., 10 steps at a time), enabling significantly more efficient inference compared to per-step prediction.
>
> | Setting    | Input Images (Resolution)       | Action Chunk Size | Inference Time | Total Latency | Action Steps/sec |
> |------------|--------------------------------|-----------------|----------------|---------------|----------------|
> | Simulation(calvin) | Third-person: 200×200, Gripper: 80×80 | 10 steps        | 1.8 s            | 1.8 s           | 5.5              |
> | Real Robot | 3× 256×256                     | 20 steps        | 2.1 s            | 3.2 s           | 6.3            |
> | Real Robot, w/ vllm | 3× 256×256                     | 20 steps        | 1.5 s            | 2.6 s           | 7.7            |
>
> For the simulation experiments, we perform inference using an NVIDIA A100 GPU (40GB). Taking CALVIN as an example, the model takes a third-person view image (200×200) and a gripper image (80×80) as input, with each inference taking an average of 1.8 seconds and outputting action chunks consisting of 10 steps. Compared with previous methods that generate one step of action per inference, our method can generate 10 steps of action per inference, which can significantly reduce computational resources.
>
> For real-world experiments, we perform model inference remotely via network communication using an NVIDIA A100 GPU (40GB). The dual-arm robot platform (AgileX Cobot Magic) takes three image inputs, each with a resolution of 256×256. The action is predicted in chunks of 20 steps, corresponding to approximately 3.3 seconds of motion window. Each model inference takes around 2.1 seconds, and including communication and data I/O, the total latency is about 3.2 seconds — meaning the system infers 20 action steps every 3.2 seconds. These predicted actions are then executed step-by-step over 20 timesteps, with each step further interpolated 5 times to enable finer and smoother control.
>
> Moreover, our fully discrete autoregressive paradigm can seamlessly leverage acceleration techniques developed for large language models, such as vLLM-based inference, enabling efficient deployment. Specifically, vLLM can achieve approximately a **1.5x speedup in inference time**.
>
>
> **W2: real-world experiments.**
>
> We fully agree that comparing against established baselines on real-world tasks is crucial for a comprehensive evaluation. To this end, we benchmarked UniVLA on the pouring task against Pi-0, one of the strongest publicly available baselines. As shown in Table A, UniVLA achieves competitive performance overall. Pi-0 attains a slightly higher success rate on the grasp stage, while both methods show comparable performance on the pouring stage. All results are averaged over 8 trials. We observe that both methods **struggle with accurate pouring and often fail to pour precisely**.
>
>
> | Method                | Right-hand Grasp Bottle | Left-hand Grasp Cup | Pour Water | Overall |
> |------------------------|--------------------------|-----------------------|------------|---------|
> | pi-0  | 87.5                     | 75.0                   | 37.5        | 37.5    |
> | UniVLA   | 87.5                     | 62.5                  | 37.5       | 37.5    |
>
> Currently, our experiments on real-world tasks are preliminary. We will add these experiments to the main paper. This paper primarily aims to validate the core idea and framework in simulation, and we plan to further explore more real-world tasks in future work.

---

> ### Author Response · Authors · 2025-11-20
> **PART2**
>
> **W3: Performance gain from emu3.**
>
> EMU3 provides foundational capabilities for image and text modalities, similar to the VLM backbone used in OpenVLA and Pi-FAST, though its inherent understanding capabilities are actually weaker than standard VLMs. As shown in **Table 4**, **naively applying this framework without our proposed world-model post-training results in performance far below existing methods**, indicating that the gains are not solely due to the pretraining backbone. The significant improvements in our work arise from the combination of **world model-based post-training** and video-centric task modeling, which together enable the framework to achieve substantial enhancements.
>
> Importantly, the fully discrete token framework we employ serves as a simple yet effective mechanism for supporting world model training. This paradigm is not restricted to EMU3 and could potentially be applied to other backbones—for instance, by adding a diffusion decoder to Qwen-VL to supervise image prediction. In contrast to the Pi-series, which lack large-scale video pretraining, our method explicitly incorporates **video-based pretraining**, emphasizing a central contribution of our framework.
>
> **Q1: Different action encoding strategies.**
>
> We adopt action discretization to provide a consistent and unified modeling paradigm. Regarding action modeling strategies, as summarized in the table below, we compared OpenVLA’s original bin-based approach with the FAST-based approach. FAST consistently demonstrates clear advantages in both performance and encoding efficiency. We also experimented with flow-matching for action modeling, but found that both its performance and training efficiency are inferior to FAST within our framework.
>
> | Action Modeling Method | calvin ABCD |
> |-----------------------|------------|
> | OpenVLA bin-based     |   3.98     |
> | FAST DCT-based            |   4.26     |
> | Flow matching continous | 3.36  |
>
> Current experiments are conducted on the single-frame setting of the CALVIN benchmark, with $8,000$ training steps. We observe that the Flow Matching component exhibits slower convergence.
>
> **Q2：About spatial grounding task.**
>
> Spatial grounding is included primarily to demonstrate the framework’s **compatibility and potential**. This exploration is preliminary and not the main focus of the paper. Our aim is simply to verify that EMU3 can acquire basic spatial reasoning abilities—similar to models like Qwen-VL—when provided with appropriate supervision. For this purpose, we post-train EMU3 using COCO images and robot datasets with bounding-box annotations generated by perception models. We will clarify this in the supplementary material, as it is not part of the core method section.
>
> **Q3: Further discussion on architecture.**
>
> Thank you for the suggestion. We will **add a discussion in the Related Work** section comparing our framework with existing VLA architectures. Specifically, we believe that PI-0.5 adopts an action expert primarily to optimize inference efficiency in real-world deployment. In principle, such idea (action expert) could also be integrated into our framework.
>
> Our comparison **focuses on the first-stage multimodal training with PI-0.5**. The advantage of our framework lies in **video-centric modeling**, which enables richer supervision for the vision tokens and supports effective world-model post-training. Of course, there is currently a significant gap in data usage; the primary goal of this work is to demonstrate the feasibility of the video-pretraining paradigm.
>
> Regarding the question of predicting actions solely with a VLM versus adopting a dual-system–style architecture, we suggest that from an engineering perspective, a dual-system approach remains more practical. It not only improves inference efficiency but also enables asynchronous operation, allowing task decomposition and action execution to be decoupled. However, from the standpoint of multimodal learning and large-scale pretraining, we believe that a unified single-model architecture is ultimately more desirable, as it can better leverage cross-modal representations and scale more effectively.

---

### Official Review · Reviewer_nxe1 · 2025-11-01

**Soundness:** 3
**Presentation:** 3
**Contribution:** 1
**Rating:** 6
**Confidence:** 4

**Summary:**

In this paper, the authors aim to demonstrate that existing VLA methods with language-centric paradigm have difficulty in learning of cross-modal represemtations, temporal and causal dependencies across the perception-action loop. The authors introduce UniVLA, which jointly models vision, language, and action through autoregressive sequence learning. The effectiveness of the method is validated through experiments in both simulation and real-world settings.

**Strengths:**

1. This paper proposes a novel paradigm-UniVLA-that models all three modalities as discrete tokens within a shared autoregressive sequence. This represents a significant conceptual advance beyond traditional late-fusion or modality-specific architectures, enabling deeper cross-modal interaction and joint representation learning.

2. The proposed world-model post-training via video-based supervision substantially enhances temporal reasoning and data efficiency, demonstrating a clear methodological contribution.

3. The model achieves state-of-the-art performance across multiple simulation benchmarks and demonstrates strong generalization to long-horizon tasks.

**Weaknesses:**

1. Although the simulation results are impressive, the study lacks a comparison against established baseline models on real-world tasks. Robustness under noisy sensory inputs remain insufficiently validated.

2. Why does employing action prediction within this framework during post-training adversely affect model performance?

3. The 8.5B-parameter model exhibits significant latency in real-world deployment (evidenced by robotic arm stuttering in the video), critically impairing its capacity for real-time temporal information processing. Can this framework be successfully applied to other VLM with fewer parameters?

**Questions:**

N/A

---

> ### Author Response · Authors · 2025-11-20
>
> We thank the reviewer for the valuable comments and suggestions, and address each point in detail below.
>
> **W1: About real-world experiment.**
>
> We fully agree that comparing against established baselines on real-world tasks is crucial for a comprehensive evaluation. To this end, we **benchmarked UniVLA on the pouring task against Pi-0**, one of the strongest publicly available baselines. As shown in Table A, UniVLA achieves competitive performance overall. Pi-0 attains a slightly higher success rate on the grasp stage, while both methods show comparable performance on the pouring stage. All results are averaged over 8 trials. We observe that both methods still struggle with accurate pouring and often fail to pour precisely.
>
>
> | Setting                | Right-hand Grasp Bottle | Left-hand Grasp Cup | Pour Water | Overall |
> |------------------------|--------------------------|-----------------------|------------|---------|
> | pi-0  | 87.5                     | 75.0                   | 37.5        | 37.5    |
> | UniVLA   | 87.5                     | 62.5                  | 37.5       | 37.5    |
>
> In addition, to assess robustness, we further evaluate the model’s success rate in grasping cups under varying lighting conditions and altered backgrounds. As shown in Table below, UniVLA maintains stable performance under these distribution shifts. A noisy background has a greater impact, mainly because the training dataset was collected with plain white backgrounds. Removing the whiteboard during inference leads to a certain drop in performance.
>
> | Setting                 | Success Rate |
> |-------------------------|-------------|
> | Normal                  |      87.5       |
> | Varying Light           |      87.5       |
> | Noisy Background        |      75.0       |
>
> We will add these experiments to the main paper. This paper primarily aims to validate the core idea and framework in simulation, and we plan to further explore more real-world tasks in future work.
>
> **W2: Explanation of performance with action prediction.**
>
> In Table 4, we observe that post-training solely on action prediction leads to a degradation in the model’s general performance. We believe this is primarily due to the **non-uniformity of the action space across different robot datasets**. Although we follow the OpenVLA action process, difference in robot embodiments, control frequencies, and action normalization procedures result in heterogeneous action distributions, which negatively impact generalization.
>
> We believe that a more unified definition of the action space would likely yield performance gains. However, our experiments reveal a more important insight: **compared with action-only post-training, vision-based supervision offers significantly better generalization** and does not depend on action labels, enabling the use of much larger quantities of video data. Moreover, we find that combining vision and action supervision yields performance comparable to vision-only post-training.
> Therefore, we adopt vision-based world-model supervision as the primary post-training objective, given its **ability to leverage substantially larger amounts of unlabeled video data**.
>
> **W3: Latency and other VLMs.**
>
> Indeed, since our engineering optimizations are not yet mature and the code was run directly on real hardware, the latency is relatively high, which results in visible jitter.
> We anticipate that with advances in inference optimization techniques, such as vllm, these issues can be mitigated in future deployments. Specifically, our method achieves a proven 1.5x speedup when implemented using vllm. For example, recent embodied models such as **Gen-0** demonstrate that even **10B-scale** models can achieve smooth, real-time responses.
>
> Moreover, our fully discrete framework is flexible and can support the training of smaller models; we chose EMU3 for ease of experimentation. Importantly, the post-training world model paradigm proposed in this work is general and can be transferred to other smaller VLMs, for example, by attaching a diffusion head to Qwen2.5-VL to provide supervision for image latents.

---

### Official Review · Reviewer_U4v1 · 2025-11-09

**Soundness:** 3
**Presentation:** 3
**Contribution:** 3
**Rating:** 6
**Confidence:** 4

**Summary:**

This paper proposes a unified Vision-Language-Action (VLA) model that discretizes all modality data into tokens. Unlike previous VLAs that predict actions solely from current observations, this work predicts the next actions conditioned on interleaved visual-action histories. Furthermore, the paper introduces a world-model post-training strategy to effectively transfer knowledge from pretrained multimodal models to downstream robotic tasks. Extensive experiments across multiple simulation environments validate the effectiveness of the proposed approach.

**Strengths:**

1. The paper presents a unified VLA framework that discretizes all modalities into tokens, resulting in a shared representation across visual, textual, and action modalities.
2. Comprehensive experiments are conducted in both simulated and real-world scenarios, validating the effectiveness of the proposed framework and analyzing the transferability of different visual post-training strategies.
3. The model is further evaluated beyond robotic manipulation, including autonomous driving tasks, demonstrating the generalization ability of the proposed approach.

**Weaknesses:**

1. Inference speed – The model is built upon Emu3 with 8.5B parameters and employs next-token prediction for action generation. This results in slow inference and limits its ability to handle tasks requiring high responsiveness. The authors are encouraged to provide detailed inference speed comparisons with other VLAs such as OpenVLA, π₀, and UVA.
2. While the paper presents a strong discrete-token–based autoregressive framework, it lacks systematic comparisons or discussions with continuous-action (e.g., π₀, DexVLA[1]) or hybrid VLA models (e.g., HybridVLA[2]). Continuous or hybrid approaches often achieve smoother control and better precision, and an in-depth analysis of these trade-offs would strengthen the paper.
3. The proposed UniVLA predicts the next chunk of actions conditioned on interleaved observation and state history, while most baseline VLAs rely only on current observations. This may create an unfair comparison. The authors should include stronger baselines that also utilize history information (e.g., Long-VLA[3]) to ensure fair comparison.


[1]. DexVLA: Vision-Language Model with Plug-In Diffusion Expert for General Robot Control
[2]. HybridVLA: Collaborative Diffusion and Autoregression in a Unified Vision-Language-Action Model
[3]. Long-VLA: Unleashing Long-Horizon Capability of Vision Language Action Model for Robot Manipulation

**Questions:**

1. Although the proposed world-model post-training strategy appears effective, it requires 50K training steps on 32 A100 GPUs over 4–5 days. Could the authors conduct additional ablation studies to analyze how this stage affects downstream performance—for instance, by reducing the size of the post-training dataset and evaluating the resulting impact?

2. In Appendix D, the authors state: “This design enables policy learning for embodied control and spatial reasoning through language output.” Could the authors further clarify how UniVLA performs spatial reasoning tasks without being explicitly trained on such datasets? Is this ability inherited from the source EMU3 model after two-stage training, or were spatial reasoning datasets incorporated during post-training?

---

> ### Author Response · Authors · 2025-11-20
> **PART1**
>
> We sincerely thank the reviewer for the insightful comments and constructive suggestions. We address each point in detail below.
>
> **W1: About inference speed.**
>
> First, we believe that 8.5B parameter is unlikely to become a bottleneck in the future. As large model infrastructure continues to advance, inference efficiency is improving rapidly. For example, recent embodied models such as Gen-0 demonstrate that even 10B-scale models can achieve smooth, real-time responses.
>
> Second, while inference is indeed a practical challenge at present, it can be substantially mitigated through **engineering-level optimization** and the use of **action-chunk prediction**, where the model forecasts a window of future actions that execute sequentially while the next chunk is inferred asynchronously.
>
> In addition, we present a detailed comparison of real-world inference efficiency against other models in the following.
>
> | setting     | model      | parameters | Model latency | Total latency |
> |-------------|------------|------------|---------|---------|
> | real-world  | pi-0       | 3.3B       |    1.1s     | 2.1s |
> | real-world  | pi-0 Fast  |  3.3B         |   1.9s      | 3.0s  |
> | real-world  | UniVLA     | 8.5B       |   2.1s      | 3.2s |
> | real-world  | UniVLA w/ vllm    | 8.5B       |   1.5s      | 2.6s |
>
> In the real-world setting, we adopt a server–client inference pipeline and measure both the pure model inference time (on an A100 server) and the end-to-end execution time on the robot. The results show that the dominant cost comes from shared system overheads (e.g., communication and perception I/O), and the model-level latency accounts for only a fraction of the total cost. Consequently, the inference time of our model is comparable to pi-0 Fast. The increase in Fast’s inference time mainly comes from generating more autoregressive tokens. However, this AR token-prediction process can be effectively **accelerated using existing LLM inference optimizations such as vLLM**; with vLLM acceleration, our method also achieves competitive latency.
>
> **W2: Discussion with continuous-action or hybrid VLA models.**
>
> Thank you for the suggestion. We will **add a discussion in the related works** to compare different structure choices.
>
> Discrete actions come with trade-offs: while they inevitably introduce some degree of information loss, they also provide clear benefits in terms of optimization stability and **convergence efficiency**. As shown in **Table 5b**, discrete actions consistently yield faster convergence and more stable optimization compared to continuous approaches such as RoboVLMs. This observation aligns with prior findings in the FAST paper, where discretization leads to up to a **5× improvement in convergence speed**.
>
> More importantly, a fully discrete architecture (UniVLA) offers clear **scalability advantages**: it avoids the auxiliary losses, extra modules, and sensitive hyperparameter tuning required by hybrid designs, resulting in a significantly simpler and more robust training pipeline.
> This simplicity is crucial for **large-scale video modeling**, where long-horizon, interleaved V–A–V–A-V-A sequences benefit from clean tokenization and stable autoregressive learning — a core contribution of our work.
>
> However, it also has some drawbacks: it relies on autoregressively predicting future action tokens, and when the number of tokens is large, its efficiency can be lower compared to directly generating actions with flow-matching action experts.
>
> **W3: Comparison with video-based VLA methods.**
>
> We thank the reviewer for the suggestion. First, we emphasize that a **video-based, interleaved manner is a key advantage and contribution of UniVLA**. This interleaved prediction conditioned on both observation and action history is particularly practical for real-world manipulation tasks.
>
> Second, in our comparisons with CALVIN in **Table 1**, both **RoboVLMs and GR-1 are also video-based methods**; we will make this explicit in the table to clarify the comparison. Additionally, we **supplement with single-frame experiments**: Table 2 shows the performance of UniVLA when using only single-frame inputs, achieving 94.8 on Libero, which remains highly competitive. Notably, video-based reasoning demonstrates greater advantages on long-horizon tasks.
>
> | Method | Mode      | SPATIAL | OBJECTS | GOAL  | long    | AVG  |
> |--------|-----------|---------|---------|-------|-------|------|
> | UniVLA | img sft   | 97.0    | 99.0    | 92.6  | 90.8  | 94.8 |
> | UniVLA | video sft | 95.4    | 98.8    | 93.6  | 94.0  | 95.5 |
>
> Regarding Long-VLA, we note that its code has not been released, and it does not report results on the standard CALVIN ABCD benchmark, making direct comparison difficult at this stage. We will include Long-VLA in our comparisons once the code and checkpoints become publicly available.

---

> ### Author Response · Authors · 2025-11-20
> **PART2**
>
> **Q1：Ablation study for world-model post-training.**
>
> Thank you for the suggestion. The current scale of video pretraining is still relatively small (622k) compared to many industrial embodied models (OXE dataset >1m), and thus does not fully exploit the potential of scaling video-based training. Our simulation results in **Table 4** further suggest that the **world model training paradigm may be more critical than the absolute data scale**. To support this, we have added additional experiments on data scale, summarized in the table below:
>
> | Data Scale  | Calvin ABCD | simpler-env |
> |------------|-------|-------|
> | ~100k       | 4.42 | 57.4 |
> | 622k       | 4.61 | 69.8 |
>
> However, we observe that the benefits of post-training tend to saturate in simulation benchmarks, suggesting that more challenging benchmarks or real-robot evaluations will be necessary to further validate scaling effects.
>
> **Q2：About spatial grounding task.**
>
> Spatial grounding is included primarily to demonstrate the framework’s **compatibility and potential**. This exploration is preliminary and not the main focus of the paper.
>
> EMU3 itself does not have spatial grounding abilities and requires post-training to acquire them. Similar to Qwen2.5-VL, which incorporates grounding data during post-training to enhance perception, we provide EMU3 with perception data for post-training. Specifically, we use RefCOCO dataset along with robot datasets containing bounding-box annotations generated by perception models to train EMU3 for basic spatial grounding capabilities. We appreciate the reviewer for pointing this out, and we will clarify this point in the appendix.

---

### Meta-Review · Area_Chair_qKSg · 2025-12-30

**Summary:**

I recommend acceptance for this paper, informed by the following reviewer feedback:

**U4v1**,  **nxe1** , and **Wn1i** all state that the *strength* is the unified VLA framework that predicts language, (discrete) actions, and video all through a unified architecture. **nxe1** and  **Wn1i** state that a *strength* is the presented evidence that world-model training effectively improves downstream policy learning.

**U4v1** also states that the *strengths* are the comprehensive experimental validation in sim and real manipulation and experiments in autonomous driving tasks. The *concerns* are that the method inference speed should be compared to other SOTA VLAs (pi0.5, openvla), lack of comparisons with continuous-action VLAs, need for fairer comparisons with how history is incorporated into the model.

**nxe1** also states that the *strengths* strong generalization on long-horizon simulation tasks. The *concerns* are about lack of baselines demonstrated in hardware and robustness, studying deeply why action prediction in post-training hurts performance, and latency.

**Wn1i** also states that the *strengths* are competitive benchmark performance. The *concerns* are about inference speed, lack of comparison of the UniVLA’s success rate on real tasks or comparing its real-world performance against other competitive VLA models, and further discussion about the model design strategy.

**6H2r** states that the *strengths* are pretraining on large-scale video prediction, consistent outperformance of prior VLA baselines, and systematic analysis of key aspects of the method. The main *concerns* are the novelty (the reviewer comments that this appears to be a “pragmatic consolidation” rather than a paradigm shift), lack of clarity in the paper, unsubstantiated claims about how the tokenization enables “causal reasoning”, lack of discussion about how discrete tokenization may fundamentally hinder performance of fine-grained tasks (hence, why other SOTA VLA architectures have moved to continuous action representations).

**Reviewer Concerns:**

**U4v1 Concerns**
* (1) inference speed should be compared to other SOTA VLAs (pi0.5, openvla): $\rightarrow$ **addressed (empirically)**
* (2) lack of comparisons with continuous-action VLAs:  $\rightarrow$ **addressed (discussion)**
* (3) need for fairer comparisons with how history is incorporated into the model:  $\rightarrow$ **not addressed**

**nxe1 Concerns**
* (1) lack of baselines demonstrated in hardware and robustness:  $\rightarrow$ **addressed (empirically, with only pi0 model baseline)**. It would be good to compare to other models mentioned by the reviewer like OpenVLA and HybridVLA.
* (2) studying deeply why action prediction in post-training hurts performance:  $\rightarrow$ **partially addressed (discussion)**
* (3) latency:  $\rightarrow$ **addressed (discussion)**

**Wn1i Concerns**
* (1) inference speed:  $\rightarrow$ **addressed (empirically)**
* (2) lack of comparison of the UniVLA’s success rate on real tasks or comparing its real-world performance against other competitive VLA models:  $\rightarrow$ **addressed (empirically, compared to pi0 on a pouring task)**
* (3) further discussion about the model design strategy:  $\rightarrow$ **addressed (discussion)**

**6H2r Concerns**
* (1) novelty: $\rightarrow$ **addressed (discussion)**
* (2) paper clarity: $\rightarrow$ **not addressed**
* (3) causality and tokenization claims:  $\rightarrow$ **addressed (discussion)**
* (4) discrete vs. continuous action representations:  $\rightarrow$ **partially addressed (discussion, some experiments).** It would be helpful to dig deeper into this and compare how the discrete action representation performs as a function of task complexity vs. SOTA models like pi0.5 that have continuous action representations.

**Reviewer Scores:**

* **U4v1** would have *maintained* a score of 6: marginally above the acceptance threshold.
* **nxe1** would have *maintained* a score of 6: marginally above the acceptance threshold.
* **Wn1i** would have *increased* to a score of 8: accept, good paper (poster)
* **6H2r** would have *maintained* a score of 8: accept, good paper (poster)

---

### Decision · Program_Chairs · 2026-01-26

Accept (Poster)